# On the "Thixotropic" Behavior of Fresh Cement Pastes

**Youssef El Bitouri * and Nathalie Azéma**

Laboratoire de Mécanique et Génie Civil, LMGC, IMT Mines Ales, University of Montpellier, CNRS, 34000 Montpellier, France
* Correspondence: youssef.elbitouri@mines-ales.fr; Tel.: +33-4-66-78-53-67

**Abstract:** Thixotropic behavior describes a time-dependent rheological behavior characterized by reversible changes. Fresh cementitious materials often require thixotropic behavior to ensure sufficient workability and proper casting without vibration. Non-thixotropic behavior induces a workability loss. Cementitious materials cannot be considered as an ideal thixotropic material due to cement hydration, which leads to irreversible changes. However, in some cases, cement paste may demonstrate thixotropic behavior during the dormant period of cement hydration. The aim of this work is to propose an approach able to quantify the contribution of cement hydration during the dormant period and to examine the conditions under which the cement paste may display thixotropic behavior. The proposed approach consists of a succession of stress growth procedures that allow the static yield stress to be measured. For an inert material, such as a calcite suspension, the structural build-up is due to the flocculation induced by attractive Van der Waals forces. This structural build-up is reversible. For cement paste, there is a significant increase in the static yield stress due to cement hydration. The addition of superplasticizer allows the thixotropic behavior to be maintained during the first hours due to its retarding effect. However, an increase in the superplasticizer dosage leads to a decrease in the magnitude of the Van der Waals forces, which can erase the thixotropic behavior.

**Keywords:** thixotropy; yield stress; cement paste; hydration; superplasticizer

## 1. Introduction

In rheology, thixotropy characterizes a time-dependent behavior [1–3]. This phenomenon, which is generally characteristic of flocculated suspensions, reflects the progressive breakdown (under a constant shear rate) of the structure formed at rest. The rheograms (shear stress as a function of shear rate) of thixotropic materials generally display a hysteresis loop. This evolution of the rheological behavior is reversible since the structural build-up occurs if the material is left at rest.

For cementitious materials, thixotropy was used to ensure proper casting and workability, especially for self-compacting concretes or printable concretes [4,5]. In addition, it allows the maintenance of workability and fluidity to be evaluated [6,7], which is very important from a practical point of view.

During the dormant or low activity period of cement hydration, the rheological behavior of cement pastes is often considered to be reversible. However, it appears that the initial structure can never be fully restored, even during this dormant period [6,8–10]. This is why cement pastes cannot be considered as typically thixotropic materials. In fact, due to the chemical evolution induced by the initial hydration reactions, the structural build-up (or breakdown) is not reversible. Roussel et al. [1] found that the structural build-up of cement pastes may be due to two origins: colloidal interactions between cement particles, which are reversible (thixotropy), and early hydrates, which form preferentially at the contact points between cement grains (irreversible). It can be noted that the irreversible changes in fresh cement paste structures can affect workability in time.

This permanent change is thus defined as workability loss.

Furthermore, the addition of a superplasticizer decreases the contribution of hydration on the structural build-up when the cement paste is left at rest [11] and thus contributes to the decrease in the workability loss. The effect of the superplasticizer can be explained by the retarding effect.

The assessment of the contributions of the reversible flocculation (thixotropy) and the irreversible chemical evolution to the structural build-up is a very interesting challenge. Different methods based on rheological measurements have been developed to assess these contributions. One of these approaches consists of the determination of the evolution of the shear stress as a function of an ascendant and descendant shear rate. The hysteresis loop, i.e., the area between the up and down curves, is an indicator of thixotropy [12–14]. Another relevant approach to assess the structural build-up is to use oscillatory measurements, such as small amplitude oscillating shear (SAOS), which allow measurements of the viscoelastic properties of suspensions (storage modulus G'; loss modulus G") within the linear viscoelastic region [1,15–18].

Another method consists of determining the evolution of the static yield stress (the minimum stress that induces flow) by a stress growth procedure [8,10,19–21]. The slope ($A_{thix}$) of the static yield versus the resting time curve is a relevant indicator of the structural build-up. In literature, this slope represents the flocculation rate and describes the reversible part of the structural build-up (thixotropy) due to particle flocculation. The order of magnitude is between 0.1 and 1.7 Pa/s [1,15].

Furthermore, the contribution of the chemical evolution during the dormant period is generally neglected, and the application of a strong shearing or remixing is considered sufficient for erasing the structural build-up. However, it appears that the structural build-up during the dormant period is not fully reversible. Recently, by using oscillatory measurements, Zhang et al. [15] found that the irreversible part of the structural build-up cannot be neglected and suggested that the structural build-up can be quantified by $A_{struct}$, which is the sum of the thixotropic part and the chemical part:

$$A_{struct} = A_{thix} + A_{chem} \tag{1}$$

However, Zhang et al. did not provide quantification of $A_{struct}$. Based on their results, $A_{struct}$ is about 0.07 Pa/s, $A_{thix}$ is about 0.06, and $A_{chem}$ is 0.01 Pa/s.

The aim of this study is to propose another approach based on the static yield stress measurement able to quantify the contribution of the chemical evolution to the structural build-up during the dormant period of cement hydration. This method is tested on ordinary Portland cement and calcite. The effect of a superplasticizer on the structural build-up is examined.

## 2. Materials

In this study, an ordinary Portland cement (CEM I 52.5 R CE CP2 NF) provided by Lafarge Holcim is used. This cement is composed of clinker (95%) and gypsum (5%). Its specific surface (Blaine) measures at 4420 $cm^2/g$, and its density is about 3.14 $g/cm^3$. In addition to cement, an inert carbonate of calcium (calcite) provided by Omya BL is used. Its density is about 2.75 $g/cm^3$, and its BET-specific surface is 2.25 $m^2/g$. Calcite is commonly used as a model material to mimic the behavior of complex cementitious materials during the dormant period [16,22–24].

The particle size distributions of the cement and calcite are determined in water using a laser granulometer (LS 13320) from Beckman Coulter Company with an adapted optical model (Figure 1). The physical properties are summarized in Table 1.

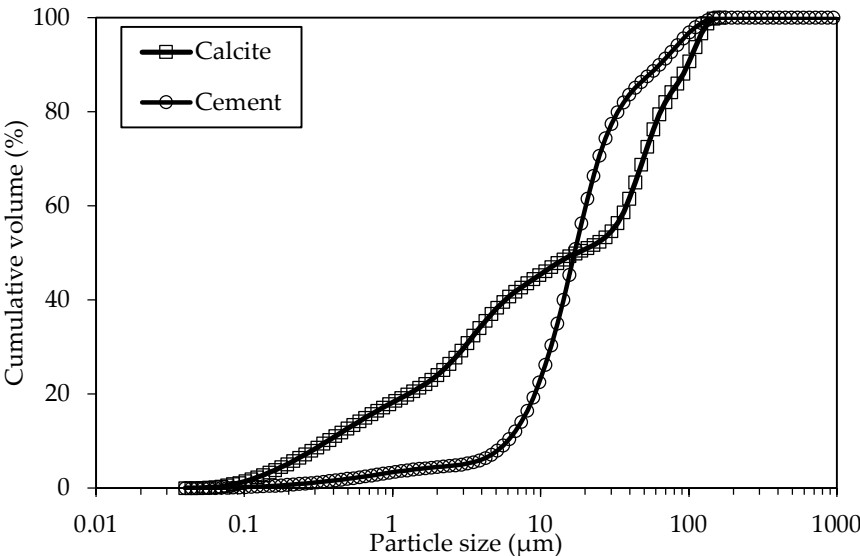

**Figure 1.** Particle size distributions of cement and calcite.

**Table 1.** Physical properties of cement and calcite.

| Parameter | Cement | Calcite |
|---|---|---|
| Mean diameter ($\mu$m) | 21.2 | 37.9 |
| $d_{10}$ ($\mu$m) | 3.9 | 0.4 |
| $d_{50}$ ($\mu$m) | 15.3 | 19.2 |
| $d_{90}$ ($\mu$m) | 45.6 | 108.4 |
| Density (g/cm$^3$) | 3.14 | 2.75 |
| Blaine-specific surface (cm$^2$/g) | 4420 | - |
| BET-specific surface (m$^2$/g) | - | 2.25 |

A commercial polycarboxylate-based superplasticizer (PCE) from Masters Builders with an equivalent dry extract content of 19.5 wt% is used.

The cement and calcite pastes were mixed with deionized water with a water-to-solid ratio (E/C) of 0.4 in a planetary agitator according to the following sequence: 5 min mixing at 500 rpm, 30 s scraping the mixer walls, and 1 min mixing at 1000 rpm. Two dosages of superplasticizer are used: 0.05 and 0.1 wt% of dry substance. A delayed addition of the superplasticizer is performed after 5 min of mixing.

The samples' preparations are performed at ambient temperature (20 °C ± 2).

### 3. Methods

#### 3.1. Rheological Measurements

The rheological measurements were carried out using a rotational rheometer AR2000Ex from TA Instruments equipped with a four-blade vane geometry. The internal diameter of this geometry is 28 mm, and the outer cup diameter is 30 mm. The resulting gap is 1 mm. The geometry constants were calibrated using the Couette analogy suggested by Aït-Kadi et al. [25].

The proposed testing method consists of a succession of stress growth measurements [19,26,27] with different resting times, as shown in Tables 2 and 3. The testing procedures begin with a strong pre-shear (100 s⁻¹) to homogenize the paste in the rheometer cup and are followed by a resting time (10 min, 20 min, and 40 min). Then, stress growth (1, 2, 3, and 4) is applied to the paste. Procedure 1 allows for measurement of the rate of increase in the static yield stress due to the total structural build-up (reversible

thixotropy and irreversible chemical evolution), while Procedure 2 (Table 3) allows for erasure of the reversible part of the structural build-up via application of a strong pre-shear before the stress growth measurements.

**Table 2.** The proposed testing method for the total structural build-up (Procedure 1).

| Time (min) | Hydration Time (min) | Procedure | Shear Rate ($s^{-1}$) | Duration (s) |
|---|---|---|---|---|
| 0.0 | 7 | Pre-shear | 100 | 30 |
| 0.5 | 7.5 | Resting time | 0 | 120 |
| 2.5 | 9.5 | Stress growth 1 | 0.01 | 180 |
| 6.0 | 13.0 | Resting time | 0 | 630 |
| 16.0 | 23.0 | Stress growth 2 | 0.01 | 180 |
| 19.5 | 26.5 | Resting time | 0 | 1230 |
| 39.5 | 46.5 | Stress growth 3 | 0.01 | 180 |
| 43.0 | 50.0 | Resting time | 0 | 2430 |
| 83.0 | 90.0 | Stress growth 4 | 0.01 | 180 |

**Table 3.** The proposed testing method for the chemical structural build-up (Procedure 2).

| Time (min) | Hydration Time (min) | Procedure | Shear Rate ($s^{-1}$) | Duration (s) |
|---|---|---|---|---|
| 0.0 | 7 | Pre-shear | 100 | 30 |
| 0.5 | 7.5 | Resting time | 0 | 120 |
| 2.5 | 9.5 | Stress growth 1 | 0.01 | 180 |
| 5.5 | 12.5 | Pre-shear | 100 | 30 |
| 6.0 | 13.0 | Resting time | 0 | 600 |
| 16.0 | 23.0 | Stress growth 2 | 0.01 | 180 |
| 19.0 | 26.0 | Pre-shear | 100 | 30 |
| 19.5 | 26.5 | Resting time | 0 | 1200 |
| 39.5 | 46.5 | Stress growth 3 | 0.01 | 180 |
| 42.5 | 49.5 | Pre-shear | 100 | 30 |
| 43.0 | 50.0 | Resting time | 0 | 2400 |
| 83.0 | 90.0 | Stress growth 4 | 0.01 | 180 |

The stress growth experiment consists of measuring the shear stress evolution under a very low constant low shear rate (0.01 $s^{-1}$). The typical stress growth curve (Figure 2 and Figures A1 and A2 -Appendix A) displays two domains. The first domain, in which the shear stress increases almost linearly with the strain until it reaches a peak, is followed by a second domain (plateau) representing the steady-state flow. The peak defines the static yield stress, which is the minimum stress required to induce the first evidence of flow. The static yield stress originates from interparticle forces and direct contacts [28] and constitutes a relevant parameter for examining the workability of cementitious materials.

The experiments are carried out in triplicate, and the average values with their standard deviation are represented.

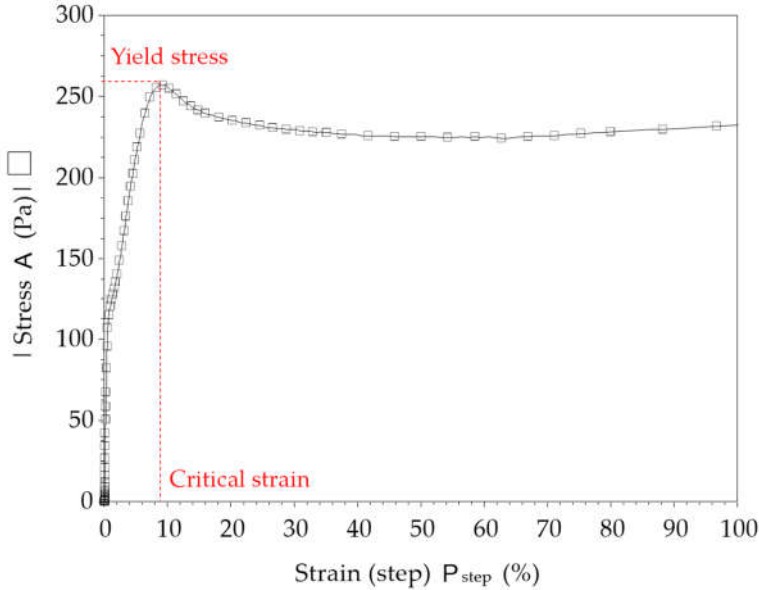

**Figure 2.** Typical evolution of shear stress under a low constant shear rate.

### 3.2. Isothermal Calorimetry

The addition of the superplasticizer leads to a retarding effect. To assess this effect, the hydration heat flow of the cement pastes is explored with an isothermal calorimeter TAM Air from TA Instruments. Pastes are prepared by external mixing at w/c = 0.4 and then introduced into the device. The calorimeter measures the difference in the heat flow between 5 g of cement paste and a reference (deionized water) at 25 °C.

## 4. Results and Discussion

### 4.1. Thixotropy vs. Non-Reversible Structural Build-Up

Thixotropy describes a time-dependent rheological behavior with reversible changes [2]. In fact, when a thixotropic material is left at rest for a long time, its yield stress (or viscosity) gradually increases. Shearing or mixing then makes it possible to recover the initial yield stress (or viscosity). At rest, there is a reversible structural build-up (flocculation) leading to an increase in the yield stress (or viscosity), whereas, under shearing, the structure formed at rest is broken (deflocculation).

For chemically inert colloidal suspensions, the structural build-up is almost reversible and is due to physicochemical interparticle interactions that lead to reversible agglomeration/dispersion phenomena. For cementitious materials, due to the chemical changes induced by hydration reactions, the structural build-up is not completely reversible; this is why they cannot be considered thixotropic materials. A part of the structural build-up induces permanent changes in fresh cement paste.

In order to assess the contribution of hydration reactions to the structural build-up, the proposed approach based on a succession of stress growth procedures is performed (Tables 2 and 3). The behavior of an inert carbonate calcium (calcite) suspension is compared to that of cement paste. The evolution of the static yield stress as a function of time is shown in Figure 3 for calcite and Figure 4 for cement paste.

For the calcite suspension, the application of a strong pre-shear before the stress growth measurements allows for the erasure of the structural build-up, as shown in Figure 3. In fact, without pre-shearing (procedure 1), the yield stress increases with time, which is characteristic of a structural build-up at rest. Moreover, the static yield stress remains almost constant with the application of shearing (procedure 2). In fact, after the first pre-shear (Table 3), the suspension is left at rest for 2 min, and then an initial static

yield stress of an order of 160 Pa is measured. A second pre-shear is applied to the suspension in order to break down the structure formed at rest, and then the suspension is left at rest again for 10 min. The second static yield stress is about 155 Pa. After resting times of 20 min and 40 min, the calcite suspension successively exhibits static yield stresses of about 156 and 143 Pa. It thus appears that a thixotropic material, such as the calcite suspension, displays a constant static yield stress that is not time dependent since the structure formed at rest is broken by the application of a strong pre-shear. The calcite suspension, therefore, represents a reference for the reversible part of the structural build-up, since no chemical reactions occur.

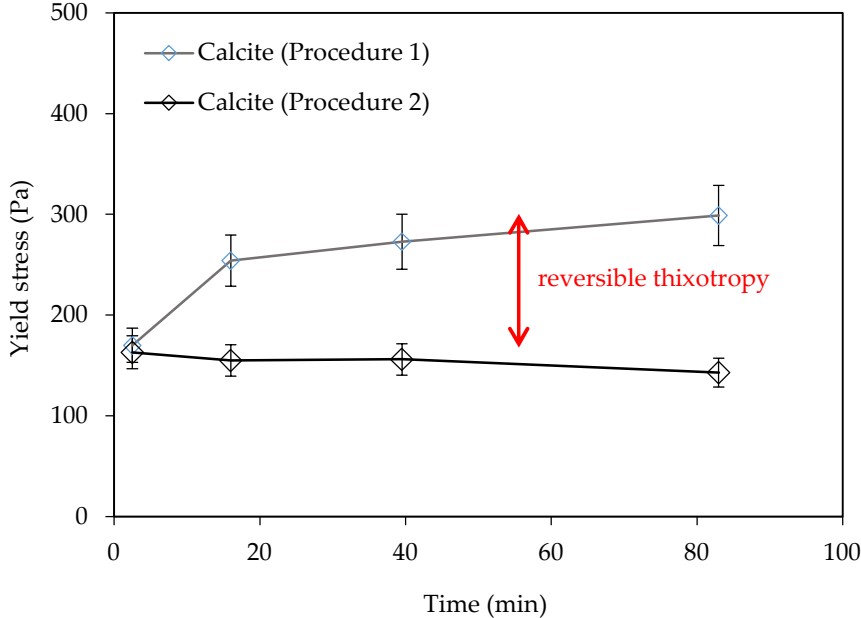

**Figure 3.** Evolution of the yield stress of the calcite suspension.

Furthermore, the static yield stress of the cement paste increases even with the application of a strong pre-shear able to erase the structural build-up. In fact, after 2 min of resting, the cement paste displays a static yield stress of 79 Pa. This static yield stress increases almost linearly with time to reach 661 Pa after 40 min of resting despite the strong pre-shear applied. This shows the irreversible nature of the structural build-up during the dormant period of cement hydration. In fact, as observed for the calcite suspension, if the cement paste is behaving as a thixotropic material, the static yield stress should remain constant with the application of the pre-shear. Figure 4 shows a significant increase in the static yield stress, which demonstrates that the irreversible structural build-up during the dormant period cannot be considered negligible.

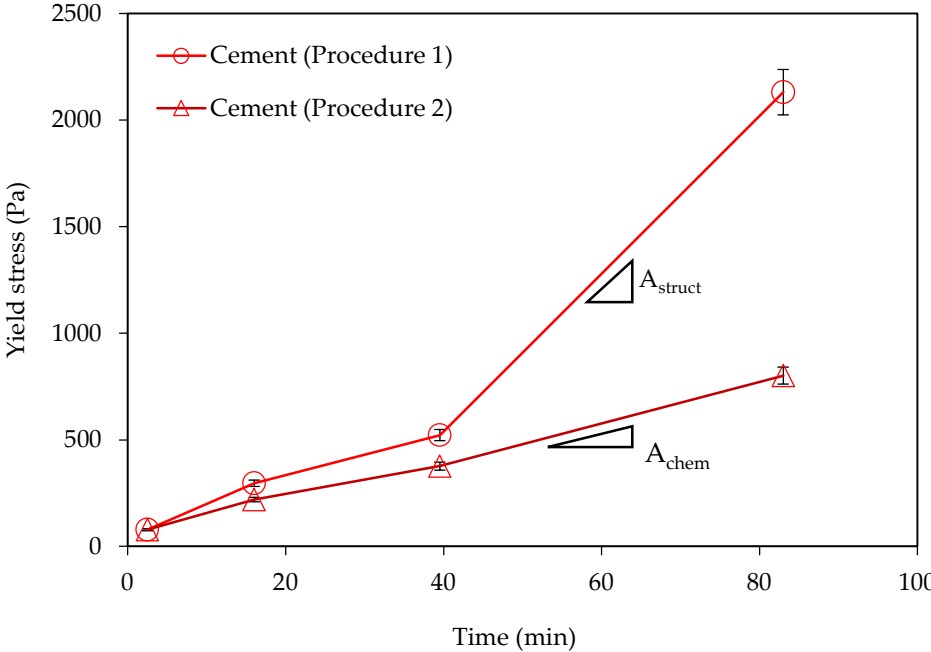

**Figure 4.** Evolution of the yield stress of the cement paste.

As suggested by Zhang et al. [15], it thus appears that the structural build-up in the cement paste is the sum of a reversible part (thixotropy) and a chemical part (early age hydration). Procedure 1 provides the total structural build-up ($A_{struct}$), while Procedure 2 allows for the evaluation of the contribution of the chemical evolution ($A_{chem}$). For a thixotropic material such as calcite, the $A_{struc}$ is equal to the $A_{thix}$ and ranges from 0.14 to 0.22 Pas/s.

### 4.2. Effect of the Superplasticizer on the Structural Build-Up

Superplasticizers are usually used to improve the workability of cementitious materials [29,30]. They adsorb onto cement particles and act by electro-steric repulsion to enhance their dispersion [31–34]. This leads to the release of water trapped between agglomerated particles. Cement paste without a superplasticizer commonly exhibits a shear-thinning behavior (i.e, a non-linear behavior with a viscosity that decreases with the shear rate), while, with the addition of a superplasticizer, the rheological behavior becomes Newtonian. In addition, the yield stress decreases with the superplasticizer dosage.

In addition to their dispersive action, superplasticizers are known to retard cement hydration [35]. The retarding effect increases with the superplasticizer dosage [36,37]. Thus, the irreversible structural build-up during the dormant period is expected to be lower than that of the cement paste without a superplasticizer.

The approach presented in Tables 2 and 3 is applied to examine the effect of the superplasticizer on the structural build-up during the first 2 h of cement hydration. The results are presented in Figure 5.

First, it can be noted that the addition of the superplasticizer leads to a decrease in the static yield stress. This effect can be explained by the dispersive action. In fact, the superplasticizer allows for the deflocculation of the cement particles via the decrease in attractive Van der Waals forces, which leads to a decrease in the yield stress.

Then, contrary to the cement paste without a superplasticizer, it can be observed that the static yield stress remains almost constant during the first hour of cement hydration for the cement paste with a superplasticizer when a strong pre-shear is applied

(Procedure 2). This indicates that the contribution of the chemical part to the structural build-up is negligible during this period. This effect may be due to the retarding effect induced by the presence of the superplasticizer, as shown in Figure 6. The static yield stress then starts increasing. Thus, the cement paste with the superplasticizer can be considered a thixotropic material during the first hour (or more, depending on the superplasticizer dosage). After this period, the contribution of cement hydration to the structural build-up cannot be neglected.

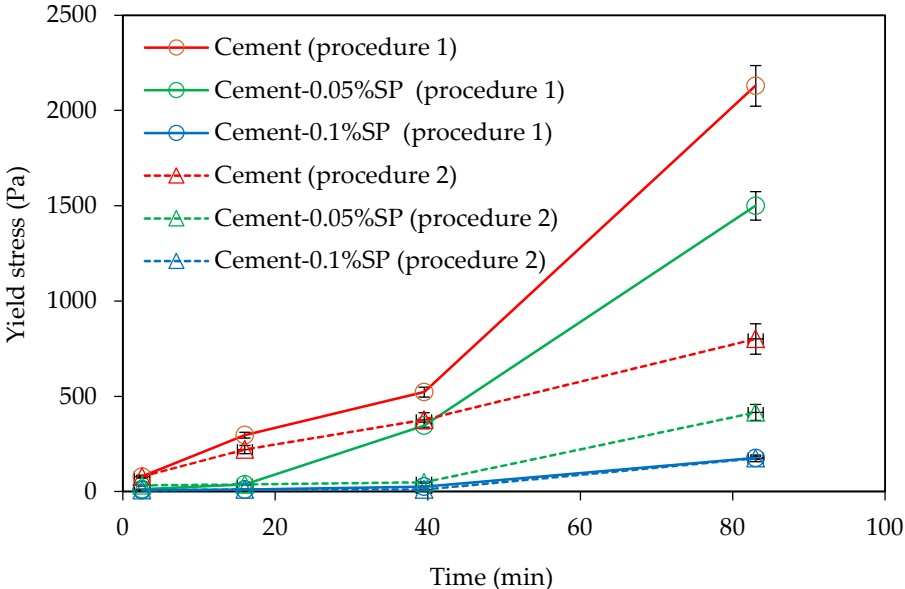

**Figure 5.** Effect of the superplasticizer dosage on the evolution of the static yield stress.

Thus, it appears that the addition of the superplasticizer affects the structural build-up during the dormant period since the superplasticizer induces a retarding effect. In this case, the contribution of cement hydration during the dormant period can be neglected. The cement paste thus behaves similarly to a thixotropic material with reversible changes. The use of the superplasticizer allows for a reduction in the workability loss during the dormant period.

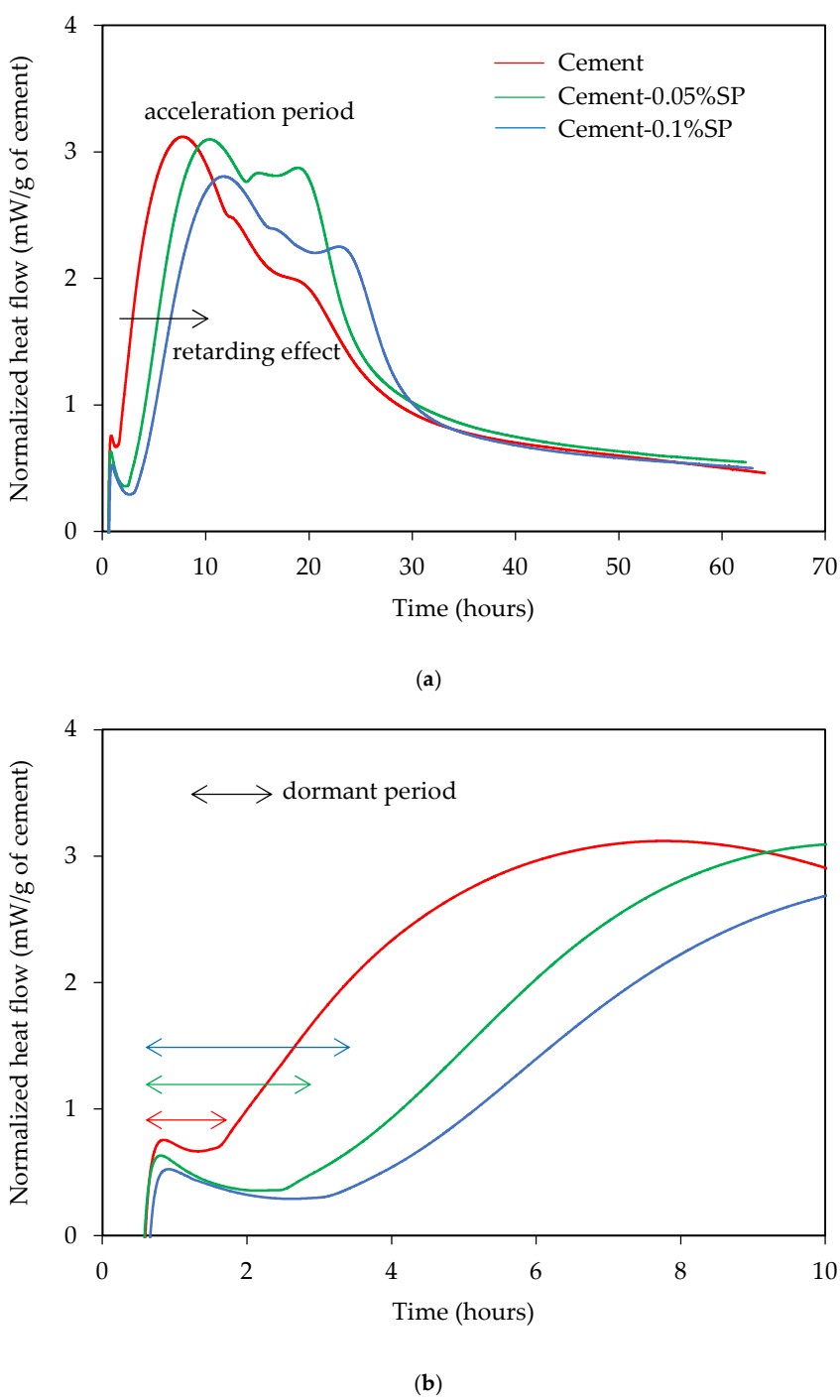

**Figure 6.** Evolution of the heat flow due to cement hydration. (**a**) retarding effect on the main peak; (**b**) retarding effect on the dormant period

The rheological procedures proposed in this work allow for quantification of the contribution of the structural build-up during the dormant period of cement hydration through the slope of the static yield stress–time curve (a derivative of the curve). As shown in Figure 7, the contribution of the irreversible chemical part ($A_{chem}$) is almost constant for the cement paste without the superplasticizer. The contribution of the reversible part ($A_{thix}$) increases with the resting time. In fact, when the cement paste is left at

rest, attractive Van der Waals forces lead to a reversible structuration. For the cement paste with 0.05% SP, both the $A_{thix}$ and $A_{chem}$ increase with time. However, due to the retarding effect (Figure 6), during the first 40 min, this paste behaves similarly to a thixotropic material since the contribution of the chemical part is negligible. Furthermore, for the cement paste with 0.1% SP, the contribution of the chemical part increases with time, while the thixotropic part remains negligible. In fact, an increase in the SP dosage leads to a decrease in the magnitude of the attractive Van der Waals forces [38,39], which are no longer able to contribute to the structuration of the cement paste at rest.

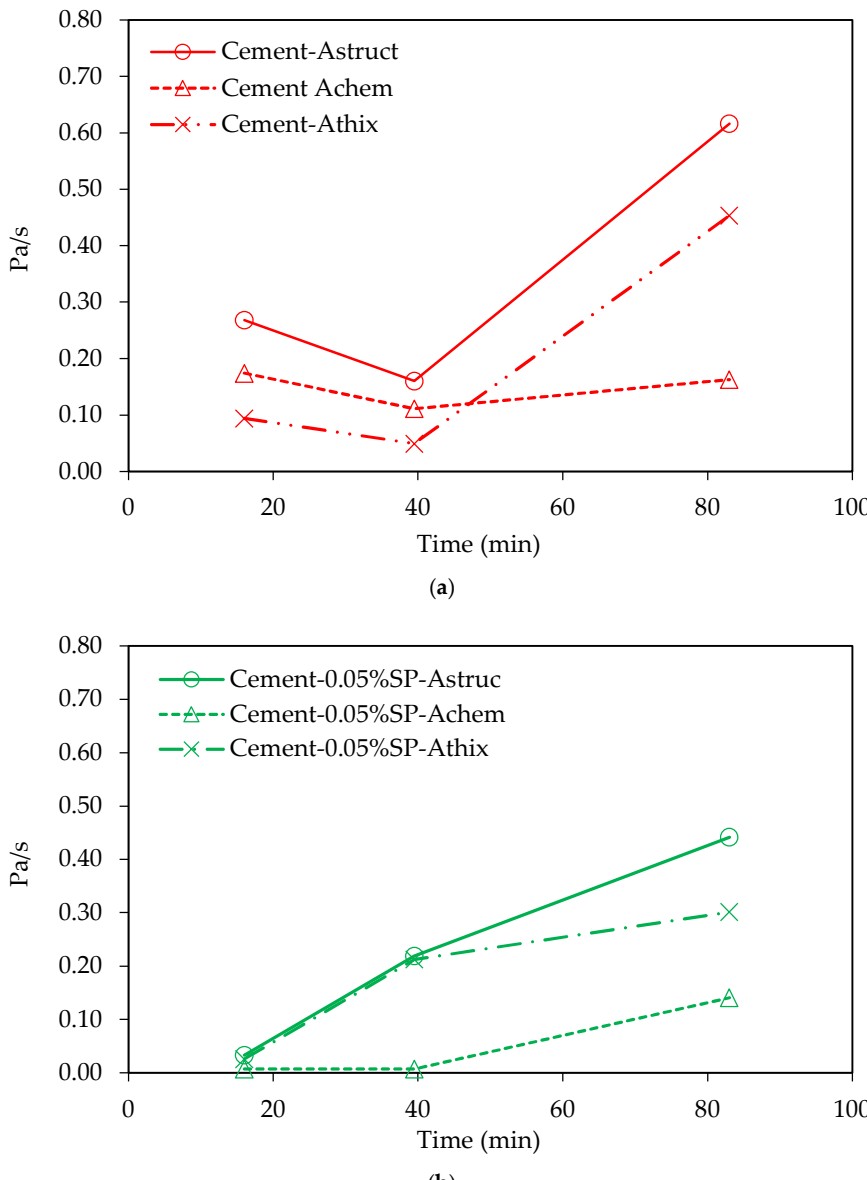

(**a**)

(**b**)

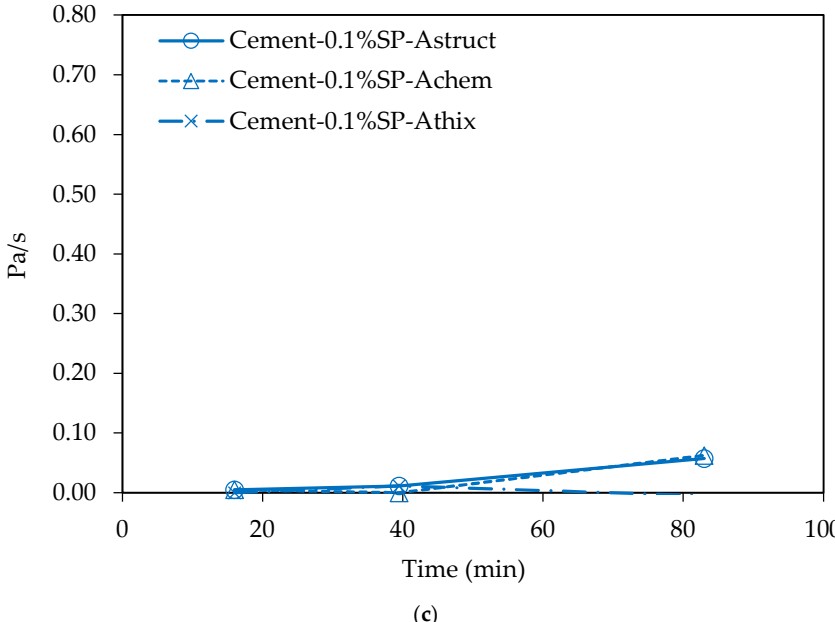

(**c**)

**Figure 7.** Evolution of the contribution of the structural build-up: (**a**) cement paste; (**b**) cement paste with 0.05% SP; (**c**) cement paste with 0.1% SP.

The approach proposed in this study is thus able to quantify the contribution of the thixotropic part and the chemical changes on the structural build-up of cementitious materials at rest. In addition, it allows the effect of the superplasticizer during the dormant period to be examined and quantified. Very few techniques make it possible to quantify the effect of the chemical changes during the dormant period. In fact, the isothermal calorimetry allows us to follow cement hydration via an indirect method (heat flow, Figure 6a), but it does not detect significant differences during the dormant period, except for its extension (Figure 6b). Combining the calorimetric data with in situ XRD patterns may detect changes during the early stage of cement hydration [40]. This proposed procedure can complete the chemical data by quantifying the rheological effect of the chemical evolution during the first hours of cement hydration.

## 5. Conclusions

In this work, an approach has been proposed to quantify the contribution of the structural build-up during the dormant period of cement hydration.

This approach has been validated on an inert thixotropic calcite suspension in which the structural build-up is almost reversible. Then, the thixotropic behavior of a cement paste without a superplasticizer has been examined. It appears that the contribution of cement hydration during the so-called "dormant period" cannot be considered negligible. In fact, there is a significant increase in the static yield stress during this period despite the strong pre-shear performed. This increase in the static yield stress describes a loss of workability that can be detrimental from a practical point of view.

Furthermore, the proposed approach allowed the effect of the superplasticizer to be investigated. It seems that the cement paste with the superplasticizer behaved similarly to a thixotropic material during the first hours of cement hydration. The structural build-up during this period can be considered reversible. Beyond this period, there are permanent changes characterized by a significant increase in the static yield stress. In addition, an increase in the superplasticizer dosage leads to a decrease in the magnitude of the attractive Van der Waals forces, which can erase the thixotropic structural build-up.

The proposed approach can thus be applied to complete the chemical data provided by other techniques, such as in situ XRD patterns and calorimetric data, to examine the chemical changes during the dormant period.

**Author Contributions:** Y.E.B.: conceptualization, methodology, validation, investigation, writing—original draft preparation. N.A.: validation, conceptualization. All authors have read and agreed to the published version of the manuscript.

**Funding:** This research received no external funding

**Conflicts of Interest:** The authors declare no conflict of interest.

**Appendix A**

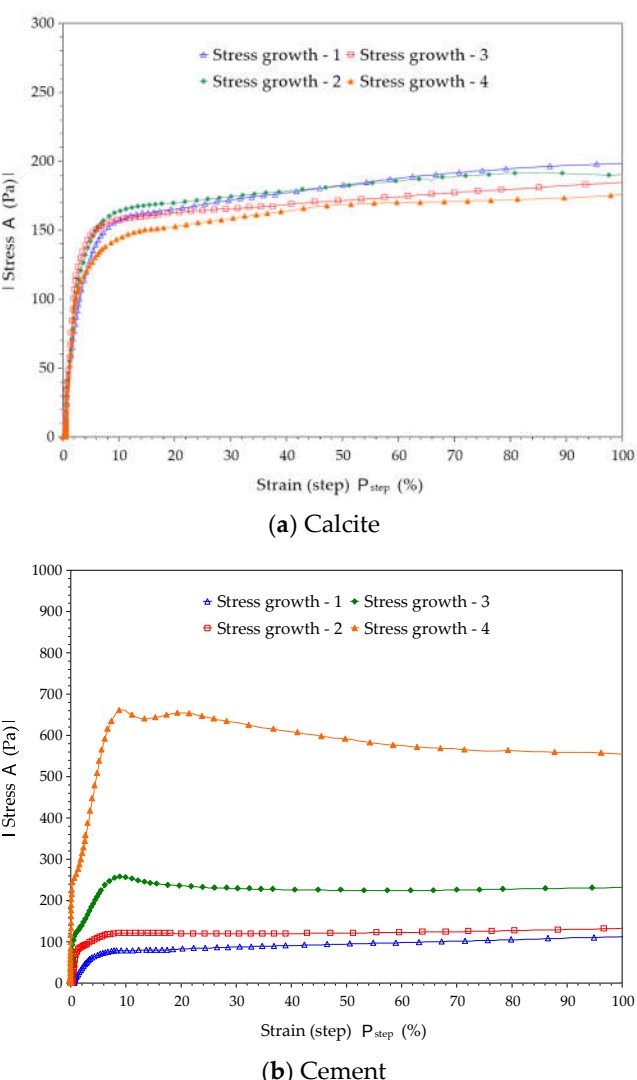

(**a**) Calcite

(**b**) Cement

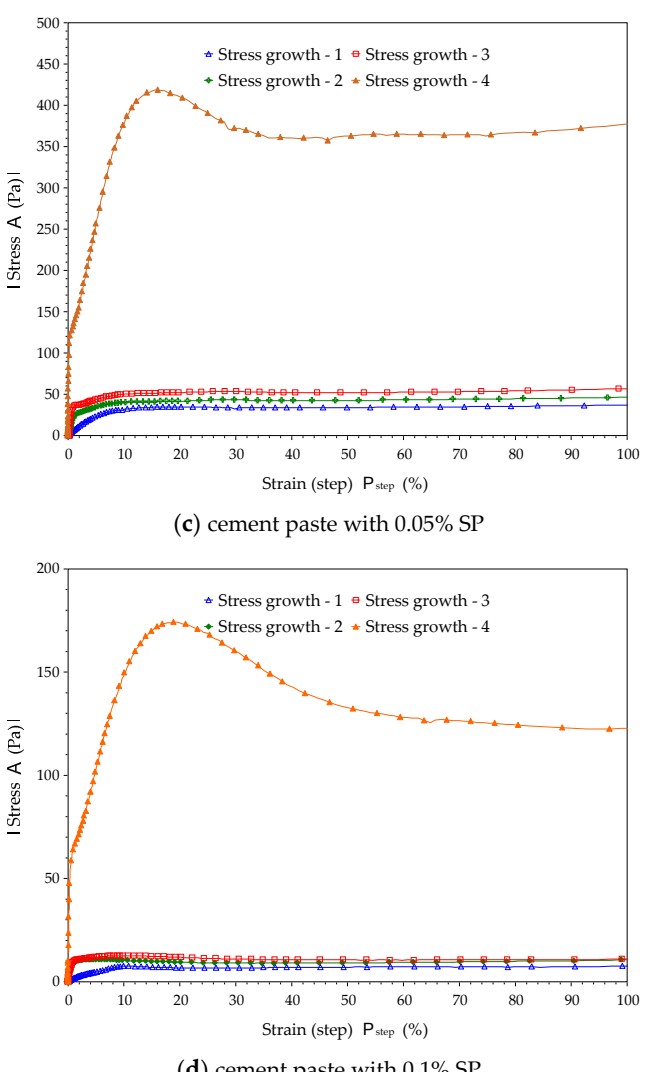

(**c**) cement paste with 0.05% SP

(**d**) cement paste with 0.1% SP

**Figure A1.** Example of the yield stress measurements of calcite (**a**), cement paste (**b**), cement paste with 0.05% SP (**c**), and cement paste with 0.1% SP (**d**) during Procedure 2.

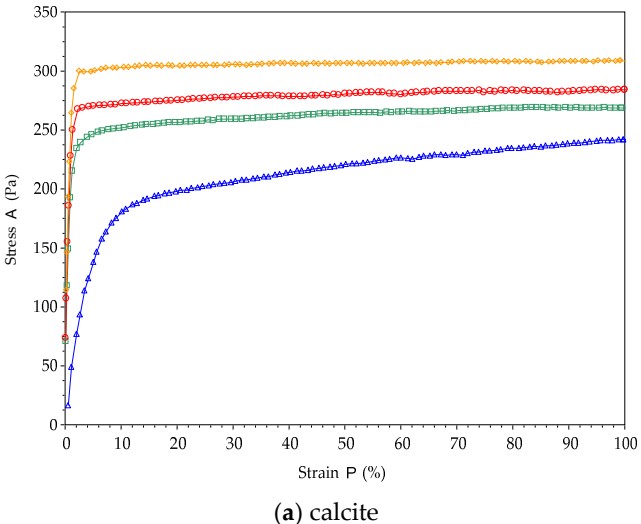

(**a**) calcite

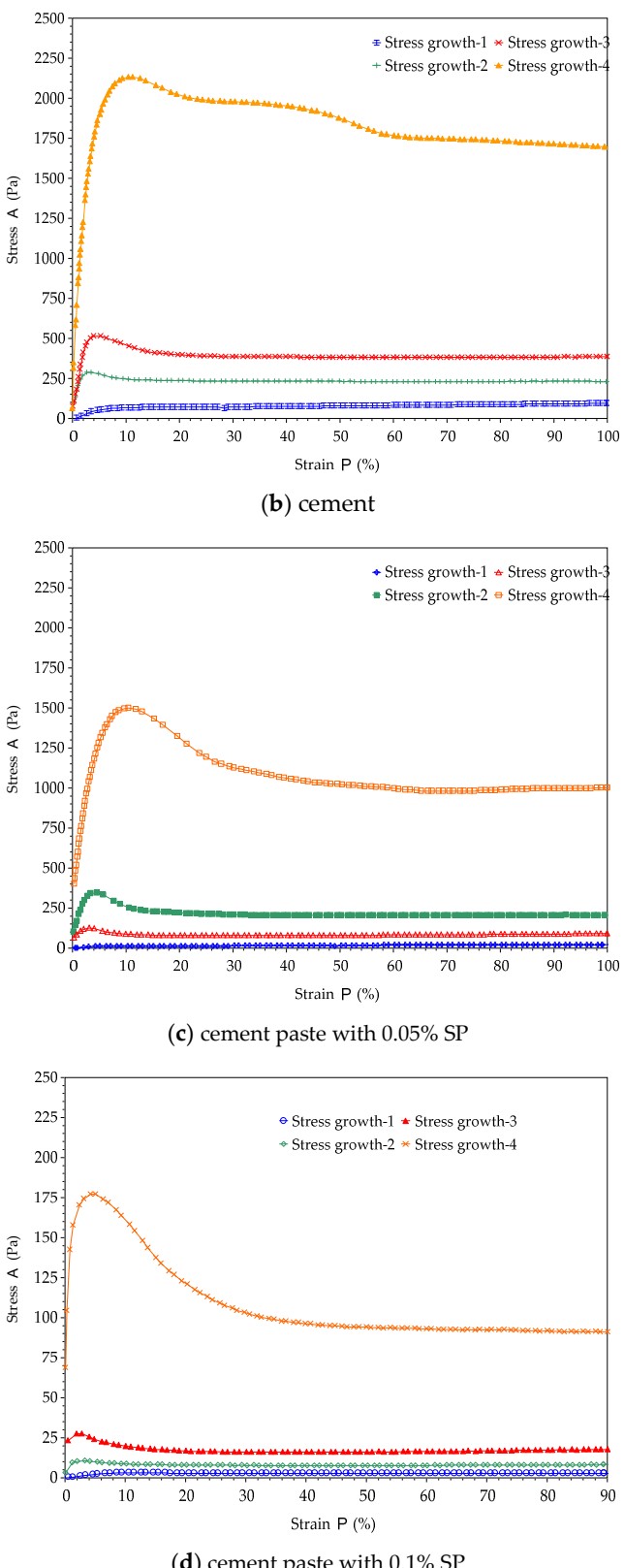

(**b**) cement

(**c**) cement paste with 0.05% SP

(**d**) cement paste with 0.1% SP

**Figure A2.** Example of the yield stress measurements of calcite (**a**), cement paste (**b**), cement paste with 0.05% SP (**c**), and cement paste with 0.1% SP (**d**) during Procedure 1.

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
