# Peer review of "On the “Thixotropic” Behavior of Fresh Cement Pastes"

_2673-4117, doi:10.3390/eng3040046_

Round 1

Reviewer 1 Report

This paper studies the influence of cement hydration behavior and superplasticizer action on the thixotropic properties of cement slurry, which is an interesting study, but the research results presented in this paper are simple and general, and lack of innovation.  Therefore, it cannot meet the requirements of publishing research academic papers.  And there is some advice in passing

(1)      Why is no specific surface area data provided for calcite? Please add

(2)      The test method in Figure 2 is not very readable, and it is best to provide a table showing the test method

Author Response

The authors would thank the reviewer 1 for his remarks and relevant comments that have made the quality of the article improved.

Below you will find the answers to the different comments made by the reviewer 1. Changes related to this comments are marked up using the “Track Changes” function of MS Word.

Reviewer 2 Report

-          The abstract does not contain information about the findings of the study.

-          Figure 2 should be checked and arranged. What is the second figure that is located on the top right?

-          What is the meaning of the square that is located on the y-axes in Figure 3.

-          When (hydration or resting time) the yield stress measurements, that are presented in Figure A1, are performed?

-          Why did not you evaluate the thixotropy by shear stress as a function of an ascendant and descendant shear rate? In this way, you can compare your approach to the other approach.

-          What is the meaning of Achem, which is calculated by yield stress improvement divided by related hydration time differences, in terms of physical explanation? 

Author Response

The authors would thank the reviewer 2 for his remarks and relevant comments that have made the quality of the article improved.

Below you will find the answers to the different comments made by the reviewer 2. Changes related to this comments are marked up using the “Track Changes” function of MS Word.

Reviewer 3 Report

This study investigated the “thixotropic” behavior of fresh cement pastes. An interesting approach has been proposed to quantify the contribution of the structural build-up during the dormant period of the cement hydration. There are some issues to clarify as follows:

Introduction

1.The lines 56-58, ‘Another relevant approach to assess the structural build-up is to use the oscillatory measurements such a as small amplitude oscillating shear (SAOS)’. Please explain what is the “small amplitude oscillating shear”?

2.The lines 61-62, ‘This method is tested on ordinary Portland cement and calcite’. Why calcite is tested? Please clarify.

Materials

3.The lines 83-85, ‘Two dosages of superplasticizer were used: 0.05 and 0.1 wt% of dry substance’. Why did the authors choose 0.05 and 0.1 wt% of dry substance?

Results

4.The lines 141-143, ‘It thus appears that a thixotropic material such as calcite suspension displays a constant static yield stress no resting-time dependent since the structure formed at rest is broken by the application of a strong pre-shear’ and the lines 143-145, ‘It has to be kept in mind, as shown in the literature, that the static yield stress of a thixotropic material should increase with increasing the resting time (Athix)’. These two sentences contradict each other, please clarify.

5.The line 170, What is ‘shear-thinning behavior’?

6. The description of Fig. 6 is brief, please describe it in more detail.

Author Response

The authors would thank the reviewer 3 for his remarks and relevant comments that have made the quality of the article improved.

Below you will find the answers to the different comments made by the reviewer 3. Changes related to this comments are marked up using the “Track Changes” function of MS Word.

Round 2

Reviewer 1 Report

Although the author has revised my opinion and replied seriously, due to the limited innovation of this article, I still reserve my opinion. I don't think this article meets the standards for publication 

Reviewer 2 Report

Thank you for your contribution